# Detection of Male Hypogonadism in Patients with Post COVID-19 Condition

**DOI:** 10.3390/jcm11071955

**Published:** 2022-03-31

**Authors:** Yukichika Yamamoto, Yuki Otsuka, Naruhiko Sunada, Kazuki Tokumasu, Yasuhiro Nakano, Hiroyuki Honda, Yasue Sakurada, Hideharu Hagiya, Yoshihisa Hanayama, Fumio Otsuka

**Affiliations:** Department of General Medicine, Okayama University Graduate School of Medicine, Dentistry and Pharmaceutical Sciences, Okayama 700-8558, Japan; grs.dg.jp@gmail.com (Y.Y.); otsuka@s.okayama-u.ac.jp (Y.O.); naru.kun.red.1117@gmail.com (N.S.); tokumasu@okayama-u.ac.jp (K.T.); me421055@s.okayama-u.ac.jp (Y.N.); hydrogen77@me.com (H.H.); sakurada202@gmail.com (Y.S.); hagiya@okayama-u.ac.jp (H.H.); hanayama@md.okayama-u.ac.jp (Y.H.)

**Keywords:** fatigue, hypogonadism, LOH syndrome, long COVID, testosterone

## Abstract

The pathogenesis and prognosis of post COVID-19 condition have remained unclear. We set up an outpatient clinic specializing in long COVID in February 2021 and we have been investigating post COVID-19 condition. Based on the results of our earlier study showing that “general fatigue” mimicking myalgic encephalomyelitis/chronic fatigue syndrome (ME/CFS) is the most common symptom in long COVID patients, a retrospective analysis was performed for 39 male patients in whom serum free testosterone (FT) levels were measured out of 61 male patients who visited our clinic. We analyzed the medical records of the patients’ backgrounds, symptoms and laboratory results. Among the 39 patients, 19 patients (48.7%) met the criteria for late-onset hypogonadism (LOH; FT < 8.5 pg/mL: LOH group) and 14 patients were under 50 years of age. A weak negative correlation was found between age and serum FT level (r = −0.301, *p* = 0.0624). Symptoms including general fatigue, anxiety, cough and hair loss were more frequent in the LOH group than in the non-LOH group (FT ≥ 8.5 pg/mL). Among various laboratory parameters, blood hemoglobin level was slightly, but significantly, lower in the LOH group. Serum level of FT was positively correlated with the levels of blood hemoglobin and serum total protein and albumin in the total population, whereas these interrelationships were blurred in the LOH group. Collectively, the results indicate that the incidence of LOH is relatively high in male patients, even young male patients, with post COVID-19 and that serum FT measurement is useful for revealing occult LOH status in patients with long COVID.

## 1. Introduction

As of February 2022, two years have passed since the declaration of the coronavirus disease 2019 (COVID-19) pandemic [1]. Vaccination has been progressing and acute treatment strategies have been established; however, the increasing number of patients suffering from various symptoms after the acute phase of COVID-19 is becoming a social problem [2,3]. These persistent sequelae, which have been termed “post COVID-19 condition” by the World Health Organization [4], commonly include general fatigue, dysgeusia, dyssomnia, low-grade fever, headache, and alopecia [5]. Notably, at least one-third of COVID-19 patients have been reported to suffer from post COVID-19 condition [2,3,6,7]. To provide comprehensive treatment and care for these patients, we established a COVID-19 aftercare clinic (CAC) at Okayama University Hospital in February 2021, and we analyzed the characteristics of patients and their clinical courses. Our previous study revealed that general fatigue recollecting myalgic encephalomyelitis/chronic fatigue syndrome (ME/CFS) was a major complaint in more than half of the patients [8,9].

The prognosis and definitive treatment for post COVID-19 condition have not been established [10]. The symptoms are thought to be associated not only with the direct effects of the virus itself on organs but also with secondary damage to the multi-organ system and physical and mental stress caused by long-term treatment and isolation due to the acute stage of COVID-19 [2,3]. Based on the clinical characteristics of post COVID-19 condition, the pathophysiology is considered to be associated with ME/CFS, post-intensive care syndrome (PICS), post-traumatic stress disorder (PTSD), or post-viral fatigue syndrome [11,12]. However, little is known about the mechanisms at molecular and cellular levels, and therapeutic targets have not been identified [13].

Recently, it has been suggested that endocrine disorders also underlie the mechanisms of post COVID-19 condition [14]. Several endocrine organs including the hypothalamus, pituitary gland, adrenal gland, thyroid gland, and genital gland have been reported to be damaged in patients with severe acute respiratory syndrome. In the present study, we focused on male hypogonadism, called late-onset hypogonadism (LOH). In patients with LOH syndrome, the impaired production and secretion of testosterone can directly deteriorate accompanying fatigue and metabolic syndrome in addition to sexual impotency [15,16]. The aim of this study was to clarify the incidence of LOH and the clinical characteristics of patients with LOH who visited our outpatient clinic.

## 2. Patients and Methods

### 2.1. Patients’ Characteristics

This study was a retrospective, observational study performed at Okayama University Hospital in Japan. Out of 61 male patients (a total of 136 patients) who visited the CAC between February 2021 and November 2021 due to symptoms that had persisted for more than one month after the acute phase of COVID-19; 39 male patients’ serum levels of free testosterone (FT) were measured due to suspicion of LOH and were included in this study. According to the FT values, the patients were categorized into two groups based on the cut-off value of 8.5 pg/mL. This cut-off value was determined in a previous study in which the mean value of FT in Japanese adults in their 20s was calculated and in which it was shown that the standard value for the diagnosis of LOH is FT of less than 8.5 pg/mL [17]. In the present study, patients with FT lower than 8.5 pg/mL were defined as the “LOH group” and other patients were defined as the “non-LOH group”.

We reviewed the medical records of the patients and obtained information about age, body mass index (BMI), duration from the onset of COVID-19 to the first visit to the clinic, acute COVID-19 treatment, and symptoms at the time of visiting the clinic. We also reviewed scores for the fatigue assessment scale (FAS), self-rating depression scale (SDS), and frequency scale for symptoms of gastroesophageal reflux disease (FSSG) at the time of the first visitation.

### 2.2. Measurement of FT and Analysis of Laboratory Data

We reviewed each patient’s laboratory results for blood cells, biochemistry and hormones including FT at the time of the first visit. The decision to measure FT level was made by each physician. Blood samples were collected in a sitting position. Information on the following biochemical parameters was also obtained: adrenocorticotropin (ACTH), cortisol, growth hormone (GH), insulin-like growth factor (IGF)-I, free thyroxin (FT4), and thyrotropin (TSH). The levels of FT and these parameters were determined by using the auto-analyzer system Cobas 8000 (F. Hoffmann-La Roche AG, Basel, Switzerland) at the Central Laboratory of Okayama University Hospital. Serum FT was determined by a radioimmunoassay (RIA; Free Testosterone RIA kit “SML”, DENIS Pharma, K.K., Tokyo, Japan) with ARC-950 (Hitachi Aloka Medical, Ltd., Tokyo, Japan).

### 2.3. Statistical Analysis

The data were analyzed by the Mann–Whitney U test, Fisher’s exact test or Pearson’s correlation coefficient to determine differences. All tests were performed as two-sided, and *p*-values less than 0.05 were regarded as statistically significant. All statistical analyses were performed using EZR, version 1.55 (Saitama Medical Center, Jichi Medical University, Saitama, Japan), a graphical user interface for R (The R Foundation for Statistical Computing, Vienna, Austria) [18].

### 2.4. Ethics

Information regarding the present study was provided on our hospital wall and on the website of our hospital, and patients who wished to opt out were offered that opportunity. Informed consent from the patients was not necessary due to the anonymization of data. This study was approved by the Ethical Committee of Okayama University Hospital (No. 2105-030) and adhered to the Declaration of Helsinki.

## 3. Results

### 3.1. Patients’ Backgrounds

Nineteen patients (48.7%) were assigned to the LOH group and the remaining 20 patients (51.3%) were assigned to the non-LOH group. Patient backgrounds including age, BMI, duration from onset of COVID-19 to the first visit, and acute COVID-19 treatment are summarized in Table 1. The median age of patients in the LOH group was 36.0 (interquartile range (IQR): 28.0–50.0) years and that of patients in the non-LOH group was 38.5 (IQR: 24.3–47.5) years. The median BMI and median duration from the onset of COVID-19 to the first visit were 25.1 (IQR: 22.1–26.5) kg/m^2^ and 71 (IQR: 49.0–105.5) days, respectively, in the LOH group and they were 23.6 (21.1–25.0) kg/m^2^ and 81 (51.3–119.5) days, respectively, in the non-LOH group. Twelve patients were provided medical care at accommodation facilities, fourteen patients were hospitalized, and the remaining sixteen patients stayed at home during the acute phase of COVID-19. There was no significant difference between the two groups in any of the patients’ backgrounds.

### 3.2. Serum FT and Correlation with Age

The median serum FT levels were 7.2 pg/mL (IQR: 5.7–7.4 pg/mL) in the LOH group and 11.0 pg/mL (9.6–14.3 pg/mL) in the non-LOH group, and the difference was statistically significant (*p* < 0.05). As shown in Figure 1A, serum FT level was negatively correlated with age (r = −0.301, *p* = 0.0624). The population of the LOH groups accounts for 46.6% (14 cases) of the 30 patients whose age was under 50 years, while that was 55.6% (5 cases) of the nine patients aged over 50 years, though the difference was not statistically significant (*p =* 0.716). Of note, the LOH group (*n* = 19) included 14 patients (74%) aged under 50 years (Figure 1B).

### 3.3. Characteristic Symptoms in the Two Groups

In medical interviews, the 39 male patients with sequelae of COVID-19 were asked about 27 symptoms. The symptoms at the time of visiting the CAC in the two groups are shown in Figure 2. The most common symptom was general fatigue, which was reported by 18 patients (94.7%) in the LOH group and by 15 patients (75.0%) in the non-LOH group. Among the 27 symptoms, general fatigue, anxiety, dysgeusia, hair loss and cough were more frequent in the LOH group than in the non-LOH group. The median scores for FAS, FSSG and SDS are shown in Table 2. The scores for FAS and SDS were not different between the LOH and non-LOH groups, while FSSG scores tended to be higher in the LOH group, though the difference was not statistically significant.

### 3.4. Serum FT and Correlations with Laboratory Parameters

A comparison of the laboratory parameters in the two groups is shown in Table 3. The median blood hemoglobin level was significantly lower in the LOH group (15.1 g/dL, IQR: 14.7–15.7 g/dL) than in the non-LOH group (15.8 g/dL, IQR: 15.4–16.7 g/dL). Correlations between FT level and laboratory parameters were comprehensively analyzed in the two groups. Blood hemoglobin (Hb), serum total protein (TP), and serum albumin (Alb) showed significant correlations with serum FT levels (Figure 3A). However, the correlations of FT levels with Hb, TP and Alb were insignificant when analyzed only in patients in the LOH group (Figure 3B).

## 4. Discussion

To the best of our knowledge, this is the first report showing the coexistence of LOH syndrome in post COVID-19 condition and their clinical relationships. We focused on the differences in symptoms and laboratory parameters between patients with LOH and those without LOH.

First, our study revealed that almost half of young patients after the acute phase of COVID-19 met the criteria of LOH syndrome. Most LOH patients are over the age of 50 years and the average age when LOH develops is in the 60s [19,20]. The prevalence of LOH in our study was lower than that in a previous study conducted in a general population (48.7% in the present study vs. 58.0% in the general population [21]). However, the prevalence of LOH in individuals under 50 years of age was almost the same (46.7% vs. 44.6%) and the prevalence of LOH in individuals over 50 years of age was lower in the present study (55.6% vs. 67.2%). These results suggest that many patients, even young patients, tend to develop LOH after the acute phase of COVID-19.

In addition, the FSSG score, which has been reported to be inversely correlated with serum FT level [21], was not significantly increased in the LOH group (median, 11) compared to that in the non-LOH group (median, 8) in post COVID-19 condition. It has been reported that gastroesophageal reflux disease (GERD) and laryngopharyngeal reflux disease can be complications of COVID-19 [22,23]. Considering that FSSG scores of non-LOH patients with long COVID were reported to be even higher than FSSG scores of non-LOH patients without COVID-19 (median of 4.79) [21], the presence of GERD-related symptoms can be a clue for a suspected underlying LOH condition in male patients.

One possible underlying condition of LOH is the prolongation of primary hypogonadism [24], which has been observed in the acute phase of COVID-19 [25,26,27,28]. It is well known that severe acute respiratory syndrome coronavirus 2 (SARS-CoV-2) targets angiotensin-converting enzyme 2 (ACE2) receptors [29], and the testis and its Leydig cells are known to express ACE2 receptors abundantly [30]. Given the fact that SARS-CoV-2 was detected in the testes of cadavers of COVID-19 patients [31], it is possible that SARS-CoV-2 infiltrates into Leydig cells, leading to hyposecretion of androgens by the testis.

Prolonged testicular damage after acute COVID-19 was shown in an animal study [32], and other endocrine organs including the hypothalamus, pituitary gland, adrenal gland, and thyroid gland are also likely to be affected by COVID-19 from the acute phase through to the chronic phase [33,34]. In addition to pulmonary and neurological damage, direct infiltration of SARS-CoV-2 into the hypothalamus or pituitary gland may activate related autoimmunity and then induce an ME/CSF-like condition through central hypogonadism [14]. From this point of view, not only primary hypogonadism but also secondary hypogonadism should be considered in post COVID-19 condition.

In the present study, we found positive correlations of serum FT levels with levels of Hb, TP, and Alb, but these correlations were blurred when focusing on the LOH group. Androgens have been reported to be associated with the production of red blood cells and serum proteins [35,36], and these positive correlations are compatible with the results of previous studies [36,37]. The results of the present study showing that the relationships between androgens and these markers became insignificant in patients in the LOH group suggests that impaired erythropoiesis in post COVID-19 condition can be, at least in part, caused by the hypogonadal condition. Since serum FT reduction is also known to be related to male frailty, the accompanying hypogonadism may also be linked to the pathogenesis of post COVID-19 condition [38] as indicated by reductions of anabolic markers.

Some studies have shown that lower serum FT levels are related to severe inflammation in patients with COVID-19 [24] and to a poor prognosis in patients with COVID-19 [28]. However, the LOH condition has not been well recognized in the aspect of post-COVID symptoms and the existence of LOH in patients with long COVID seems likely to have been overlooked. Therefore, it would be beneficial for both patients and physicians to identify LOH in post COVID-19 patients in that it may reassure the patients and it may be helpful for considering treatment strategies. In our study, some key symptoms, such as general fatigue and anxiety, both of which are typical LOH symptoms, were prominent also in LOH patients after the acute phase of COVID-19. In addition, the levels of blood Hb were lower in the LOH patients, indicating that anemia, even if the degree is modest, might be a clue for detecting LOH in long COVID patients. Although serum FT measurement is not usually included in screening tests for post COVID-19 [39,40], it would be meaningful to measure serum FT in some patients with post COVID-19 condition, especially patients who complain of fatigue and anxiety or present signs of persistent anemia.

There were some limitations in the present study. Firstly, our study was performed retrospectively at a single center with a relatively small number of patients. Additionally, we could not perform a powerful statistical analysis of the factors related to the development of LOH. Secondly, serum FT level was not measured in all of the male patients in our study, and there might have been a selection bias by the physician. The higher pre-test probability for LOH may have resulted in a higher incidence of LOH in our study than that in the general population. Thirdly, the aging males’ symptoms (AMS) rating scale was not measured and analyzed in this study. Finally, we did not investigate the clinical course, treatment and prognosis after the diagnosis of LOH.

In conclusion, our study revealed that there were many patients, even young patients, with LOH among the patients who visited our CAC. Not only primary hypogonadism but also other secondary pathogeneses related to acute infections with the SARS-CoV2 virus might have been involved in the changes of anabolic markers due to FT reduction, which may have complicated the clinical symptoms of post COVID-19 condition. It is, therefore, important to consider the necessity of measuring serum FT in patients, especially in patients with fatigue, anxiety, or signs of anemia, at an early stage of post COVID-19 condition. Further prospective investigation for treatments and prognoses is necessary for establishing the clinical significance of LOH in post-COVID condition.

## Figures and Tables

**Figure 1 jcm-11-01955-f001:**
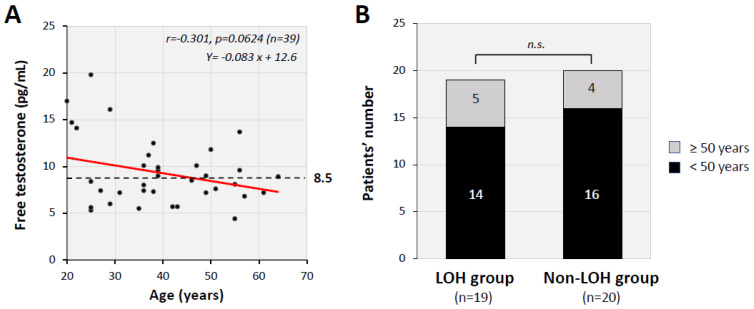
Relationship between serum FT levels and age. (**A**) Serum FT levels showed a weak negative correlation with age (r = −0.301, *p* = 0.0624). (**B**) The number of patients under 50 years of age in each group is shown. Fourteen patients (73.7%) in the LOH group were under 50 years of age and 16 patients (80%) in the non-LOH group were under 50 years of age. There was no significant difference (*p* = 0.716) by Fisher’s exact test. n.s.: not significant.

**Figure 2 jcm-11-01955-f002:**
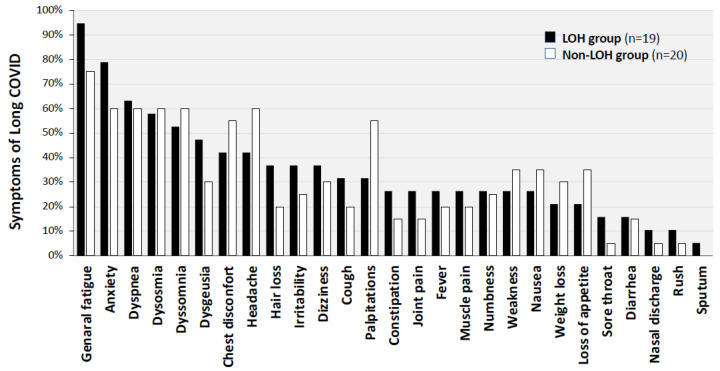
Frequent symptoms due to post COVID-19 in the LOH and non-LOH groups. The proportions of each symptom in the two groups are shown. The most common symptom in both groups was general fatigue, which was reported by 18 patients (94.7%) in the LOH group and by 15 patients (75.0%) in the non-LOH group. Fatigue, anxiety, dysgeusia, hair loss and cough were more frequent in the LOH group than in the non-LOH group.

**Figure 3 jcm-11-01955-f003:**
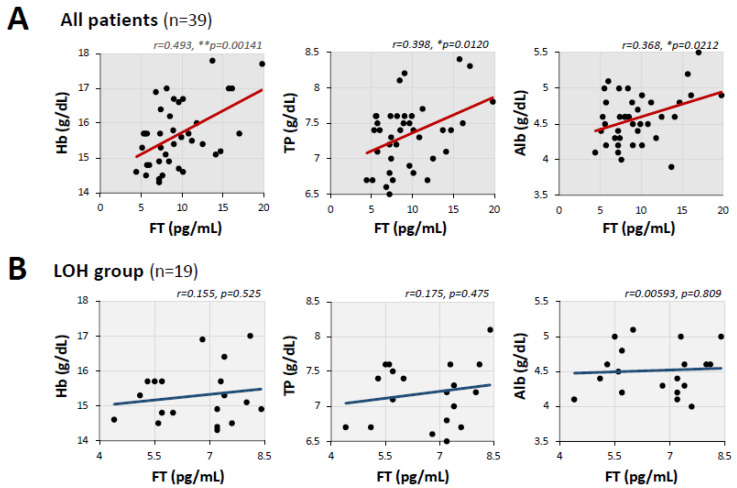
Relationship of the level of serum FT with blood level of hemoglobin and serum levels of total protein and albumin. (**A**) Blood hemoglobin (Hb), serum total protein (TP), and serum albumin (Alb) were parameters that showed significant correlations with serum FT. (**B**) These correlations of serum FT with blood Hb, serum TP, and serum Alb were not significant in the LOH group. ** *p* < 0.01 and * *p* < 0.05, statistically significant between the indicated factors.

**Table 1 jcm-11-01955-t001:** Backgrounds of patients in the LOH and non-LOH groups.

	LOH Group (*n* = 19)	Non-LOH Group (*n* = 20)	*p* Values
Age (years), median (IQR)	36.0 (28.0–50.0)	38.5 (24.3–47.5)	0.833 ^(a)^
BMI (kg/m^2^), median (IQR)	25.1 (22.1–26.5)	23.6 (21.1–25.0)	0.237 ^(a)^
Days after onset, median (IQR)	71 (49.0–105.5)	81 (51.3–119.5)	0.509 ^(a)^
Treatment of acute phase (%)			
At home	6 (31.6)	10 (50.0)	0.333 ^(b)^
At accommodation facilities	5 (26.3)	7 (35.0)	0.731 ^(b)^
Hospitalized	9 (47.4)	5 (25.0)	0.191 ^(b)^
Oxygen therapy	4 (21.1)	2 (10.0)	0.407 ^(b)^
Intensive care	0 (0)	2 (10.0)	0.487 ^(b)^

BMI, body mass index; IQR, interquartile range. The data were analyzed by (a) the Mann-Whitney U test or (b) Fisher’s exact test.

**Table 2 jcm-11-01955-t002:** Self-rating scales in the LOH and non-LOH groups.

	LOH Group (*n* = 19)	Non-LOH Group (*n* = 20)	*p* Values
FAS, median (IQR)	33 (27.5–35.5)	31 (23.0–38.0)	0.629
FSSG, median (IQR)	11 (7.5–17.0)	8 (2.25–17.50)	0.522
SDS, median (IQR)	47 (43.0–53.0)	48 (37.8–50.0)	0.692

FAS, fatigue assessment scale; FSSG, frequency scale for symptoms of gastroesophageal reflux disease; SDS, self-rating depression scale. The data were analyzed by the Mann–Whitney U test.

**Table 3 jcm-11-01955-t003:** Laboratory parameters in the LOH and non-LOH groups.

	LOH Group (*n* = 19)	Non-LOH Group (*n* = 20)	*p* Values
Blood cell counts, median (IQR)			
WBC (×10^3^/μL)	6.44 (5.18–7.98)	5.66 (4.675–7.185)	0.244
RBC (×10^6^/μL)	4.870 (4.750–5.130)	5.185 (4.875–5.308)	0.097
Hb (g/dL)	15.1 (14.7–15.7)	15.8 (15.4–16.7)	<0.05 *
Plt (×10^3^/μL)	266.0 (230.5–287.0)	242.5 (213.5–266.0)	0.227
Biochemistry, median (IQR)			
Na (mEq/L)	141 (140–141)	141 (140.0–141)	0.317
K (mEq/L)	4.2 (4.1–4.3)	4.2 (4.1–4.4)	0.864
Cl (mEq/L)	105 (104–106)	104 (103–106)	0.260
TP (g/dL)	7.2 (6.8–7.6)	7.5 (7.3–7.6)	0.078
Alb (g/dL)	4.5 (4.25–4.70)	4.6 (4.48–4.83)	0.303
T. Bil (mg/dL)	0.71 (0.57–0.96)	0.71 (0.58–0.98)	1.000
AST (U/L)	20 (18–29)	23 (20–26)	0.811
ALT (U/L)	20 (16–42)	30 (22–41)	0.266
ALP (U/L)	68 (64–87)	81 (67–105)	0.173
UN (mg/dL)	13.3 (10.5–14.5)	11.0 (10.2–14.2)	0.536
Cr (mg/dL)	0.82 (0.80–0.87)	0.78 (0.76–0.89)	0.564
LDL-C (mg/dL)	112 (97.5–138.0)	110 (85.8–146.3)	0.448
Fe (μg/dL)	97.0 (83.5–117.5)	89.5 (64.0–111.0)	0.346
CRP (mg/dL)	0.08 (0.04–0.12)	0.08 (0.03–0.19)	0.683
Ferritin (μg/mL)	223 (185.5–435.5)	250 (158.5–385.0)	0.989
BS (mg/dL)	101 (96–111)	99 (93–103)	0.448
SARS-CoV2Ab (mg/dL)	223.0 (52.0–375.0)	182.5 (59.6–5067.5)	0.607
Endocrine data, median (IQR)			
Cortisol (μg/dL)	7.8 (5.3–9.3)	7.2 (5.6–10.8)	0.922
ACTH (pg/mL)	25.5 (20.6–36.2)	28.4 (20.8–38.7)	0.588
FT4 (ng/dL)	1.3 (1.21–1.36)	1.4 (1.29–1.55)	0.148
TSH (μIU/mL)	1.21 (0.56–1.59)	1.26 (1.00–2.41)	0.164
GH (ng/mL)	0.07 (0.04–0.18)	0.14 (0.07–0.23)	0.209
IGF-I (ng/mL)	164 (131–179)	166 (136–227)	0.593
FT (pg/mL)	7.2 (5.7–7.4)	11.0 (9.6–14.3)	<0.05 *

ACTH, adrenocorticotropin; Alb, albumin; ALT, alanine aminotransferase; ALP, alkaline phosphatase; AST, aspartate aminotransferase; BS, blood sugar; Cl, chloride; Cr, creatinine; CRP, C-reactive protein; Fe, iron; FT, free testosterone; FT4, free thyroxin; GH, growth hormone, Hb, hemoglobin; IGF, insulin-like growth factor; K, potassium; LDL-C, low-density lipoprotein cholesterol; Plt, platelets; Na, sodium; RBC, red blood cells; SARS-CoV2Ab, severe acute respiratory syndrome coronavirus 2 antibody; T. Bil, total bilirubin; TSH, thyrotropin; TP, total protein; UN, urea nitrogen; WBC, white blood cells. The data were analyzed by the Mann–Whitney U test; and * *p* < 0.05, is statistically significant.

## Data Availability

Information regarding the present study was provided on our hospital wall and on the website of our hospital, and patients who wished to opt out were offered that opportunity.

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
