# Peer review of "Detection of Male Hypogonadism in Patients with Post COVID-19 Condition"

_jcm, 2022, doi:10.3390/jcm11071955_

Round 1
Reviewer 1 Report
I found this article interesting, well written and researched. The motivations are laid out and the results well presented. I think this enhances the field of long covid.
a couple of points
- i dont think you can infer that you should measure such markers in all patients now- the authors should instead say that they plan to address this prospectively and then re-present their data
- what have the authors done about the hypogonadism? i dont see anything about treatment options-could they elaborate on that?
Author Response
RE: Manuscript ID: jcm-1655167-R1
J Clin Med: Special Issue "Post-COVID Symptoms in Long-Haulers: Definition, Identification, Mechanisms, and Management"
Title: Detection of male hypogonadism in patients with post COVID-19 condition.
Authors: Yukichika Yamamoto, et al.
Dear Dr. Michaela Zheng, Editor of JCM:
Dear Ms. Ivana Zhang, Section Managing Editor:
We would like to thank the editor for the careful review and favorable comments on the manuscript. According to the editor’s and referees’ comments and questions, we revised our manuscript as addressed in the following sections. We would also like to thank you for allowing us to resubmit our revised manuscript. Also, the revised manuscript underwent English proofreading again. We hope that the revised manuscript is now acceptable for publication in JCM.
Sincerely yours,
Yukichika Yamamoto, M.D.
Fumio Otsuka, M.D., Ph.D.
Reviewer 1
Comment:
I found this article interesting, well written and researched. The motivations are laid out and the results well presented. I think this enhances the field of long covid.
… Answer: We thank you for the careful review and favorable comments on the manuscript. According to your comments and questions, we thoroughly revised our manuscript point-by-point as addressed in the following sections.
A couple of points:
1. I don’t think you can infer that you should measure such markers in all patients now- the authors should instead say that they plan to address this prospectively and then re-present their data.
… Answer: We agree with your comment. In the revised manuscript, we corrected the final parts of the abstract and the conclusion paragraph in the discussion section according to the referee’s comments.
2. What have the authors done about the hypogonadism? I don’t see anything about treatment options-could they elaborate on that?
… Answer: In the present study, we focused on detection of the LOH condition in long COVID patients. Actually, most of the treatment for LOH was performed by using Kampo medicine instead of using androgen replacement in our series of LOH patients. In a future study, we would like to clarify the treatment issue and the prognosis of LOH due to long COVID. Thank you for your important comments.
Thank you for your constructive review.
We hope that the revised manuscript is now acceptable for publication in JCM.
Reviewer 2 Report
General comments:
The authors report the frequency and correlates of low free testosterone plasmatic levels (LOH) in a retrospective cohort of 39 men attending a post-Covid19 clinic. Frequency of LOH is relatively similar to other series in the general population, but more frequent in subjects <50 years: LOH was detected in 14/30 patients <50 years, and 5/9 patients >50 years.
Contrary to the authors claim, hypogonadism has been already been reported in the context of post-Covid 19 condition, and it is necessary to update these references (for example Rastrelli et col., Endocrinology 2021, Apaydin et col., Endocrinology 2022, ...)
Patients are well described and analyzed. Correlates of LOH are classical ones rather than Covid-19 related items and this has to be more discussed by authors.
Limitations of the study should be more detailed, especially the retrospective status of the study, the low number of patients analyzed, without exhaustivity of patients seen in the care-center, not allowing a powerful statistical analysis of factors related to the characteristics of the Covid19 disease in patients with LOH. Thereby, the weakness of statistical analysis should be emphasized.
Specific comments:
- in the abstract, please reformulate the sentance: " and 14 patients (46.7%) were under 50 years of age", which falsely suggests that 46% of patients are under 50.
- add some references more of LOH after Covid19
Author Response
RE: Manuscript ID: jcm-1655167-R1
J Clin Med: Special Issue "Post-COVID Symptoms in Long-Haulers: Definition, Identification, Mechanisms, and Management"
Title: Detection of male hypogonadism in patients with post COVID-19 condition.
Authors: Yukichika Yamamoto, et al.
Dear Dr. Michaela Zheng, Editor of JCM:
Dear Ms. Ivana Zhang, Section Managing Editor:
We would like to thank the editor for the careful review and favorable comments on the manuscript. According to the editor’s and referees’ comments and questions, we revised our manuscript as addressed in the following sections. We would also like to thank you for allowing us to resubmit our revised manuscript. Also, the revised manuscript underwent English proofreading again. We hope that the revised manuscript is now acceptable for publication in JCM.
Sincerely yours,
Yukichika Yamamoto, M.D.
Fumio Otsuka, M.D., Ph.D.
Reviewer 2
General comment:
The authors report the frequency and correlates of low free testosterone plasmatic levels (LOH) in a retrospective cohort of 39 men attending a post-Covid19 clinic. Frequency of LOH is relatively similar to other series in the general population, but more frequent in subjects <50 years: LOH was detected in 14/30 patients <50 years, and 5/9 patients >50 years.
Contrary to the authors claim, hypogonadism has been already been reported in the context of post-Covid 19 condition, and it is necessary to update these references (for example Rastrelli et col., Endocrinology 2021, Apaydin et col., Endocrinology 2022, ...)
Patients are well described and analyzed. Correlates of LOH are classical ones rather than Covid-19 related items and this has to be more discussed by authors.
Limitations of the study should be more detailed, especially the retrospective status of the study, the low number of patients analyzed, without exhaustivity of patients seen in the care-center, not allowing a powerful statistical analysis of factors related to the characteristics of the Covid19 disease in patients with LOH. Thereby, the weakness of statistical analysis should be emphasized.
… Answer: We are grateful for the careful review and favorable comments on our manuscript. According to the referee’s comments and questions, we thoroughly revised our manuscript as follows:
As for the references the referee suggested, we consider that the related articles published by the suggested authors in Andrology (PMID: 32436355 & 34994082) are very suitable in our paper. Those studies have demonstrated lowered testosterone levels in patients with COVID-19 and prolongation of acute symptoms. We added those references in the revised manuscript.
Also, as the referee pointed, LOH is a well-established condition now, but the occurrence of LOH has yet to be well recognized in the aspect of post-COVID condition. Although some previous studies have shown that SARS-CoV-2 can impair testicular functions and may cause hypogonadism, the existence of LOH in patients with long COVID seems likely to be overlooked. Our report will be helpful for diagnosing LOH to focus on the key clinical signs for the detection of latent LOH status. We addressed these contents in the revised discussion.
Regarding the limitation factors in this study, we also added a detailed explanation in the revised manuscript.
Specific comment:
in the abstract, please reformulate the sentence: "and 14 patients (46.7%) were under 50 years of age", which falsely suggests that 46% of patients are under 50.
add some references more of LOH after Covid19
… Answer: We agree. We removed the percentage data of (“46.7%”) to avoid confusion. Although there are some reports of COVID-induced hypogonadism, there have been few reports indicating the existence of LOH as a post-COVID condition. We added some related papers regarding the LOH condition after COVID-19.
Thank you for your constructive review.
We hope that the revised manuscript is now acceptable for publication in JCM.